# Demonstration of Autonomous Emittance Characterization at the Argonne Wakefield Accelerator

Ryan Roussel [1,*], Dylan Kennedy [1], Auralee Edelen [1], Seongyeol Kim [2], Eric Wisniewski [2] and John Power [2]

1   SLAC National Accelerator Laboratory, Menlo Park, CA 94025, USA; kennedy1@slac.stanford.edu (D.K.);
    edelen@slac.stanford.edu (A.E.)
2   Argonne National Laboratory, Lemont, IL 60439, USA; seongyeol.kim@anl.gov (S.K.);
    ewisniew@anl.gov (E.W.); jp@anl.gov (J.P.)
*   Correspondence: rroussel@slac.stanford.edu

**Abstract:** Transverse beam emittance plays a key role in the performance of high-brightness accelerators. Characterizing beam emittance is often carried out using a quadrupole scan, which fits beam matrix elements to experimental measurements using first-order beam dynamics. Despite its simplicity at face value, this procedure is difficult to automate due to practical limitations. Key issues that must be addressed include maintaining beam size measurement validity by keeping beams within the radius of diagnostic screens, ensuring that measurement fitting produces physically valid results, and accurately characterizing emittance uncertainty. We describe a demonstration of the Bayesian exploration technique towards solving this problem at the Argonne Wakefield Accelerator, enabling a turn-key, autonomous quadrupole scan tool that can be used to quickly measure beam emittances at various locations in accelerators with limited operator input.

**Keywords:** emittance; autonomous accelerator; diagnostics

## 1. Introduction

Particle accelerators are complex instruments that require constant operator supervision and control to produce high-quality beams for use in a variety of scientific endeavors. Often, this requires measuring beam attributes that have a high impact on accelerator applications, most notably, the transverse beam emittance. A common method of measuring beam emittances is a quadrupole scan [1], where a quadrupole is used to rotate the transverse beam distribution in phase space while measuring the projected beam size on a downstream diagnostic screen.

Quadrupole scans are relatively straightforward to perform manually or automatically given prior measurements or knowledge of beam properties in the accelerator. The quadrupole strength is scanned at fixed intervals between upper and lower bounds, predetermined by operators based on prior experience or beam dynamics simulations. Quadrupole focusing strengths must be chosen such that the beam remains within the confines of the diagnostic screen and is focused enough to be resolvable above background noise in screen measurements in order to guarantee that measurements of the beam size are accurate. On the other hand, a wide range of focusing strengths must be used to sample multiple phase advances in order to accurately calculate the beam emittance.

This beam size sampling method works well for repeated measurements of beam emittances in well-understood beamline configurations. However, it becomes inefficient to perform quadrupole scans in novel contexts such as varying operational conditions or new beamlines. Determining the sample spacing and the lower and upper bounds of fixed quadrupole scans is a tedious and inefficient trial-and-error process that must be repeated for each beamline and operating configuration. As a result, it is challenging to use quadrupole scans for emittance measurements when performing optimization of upstream beamline parameters. This is especially true if upstream beamline parameters significantly

affect the beam size and divergence at the quadrupole scan location, which in turn alters the range of quadrupole strengths that lead to valid beam size measurements.

These challenges also present a barrier towards future autonomous operation of accelerator facilities. Emittance measurements using the quadrupole scan method require substantial operator oversight to configure, monitor and validate the results. Beam size measurements are often subject to errors due to noise and uncertainties. Reconstructions of the beam phase space distribution using least-squares fitting of experimental data can be strongly influenced by these errors. Even small errors in the determination of beam matrix elements $< x^2 >, < x'^2 >, < xx' >$ from this fitting can have major ramifications for calculating the beam emittances $\varepsilon = \sqrt{< x^2 >< x'^2 > - < xx' >^2}$ due to catastrophic cancellation effects. For example, if the true transverse phase space has the second-order beam moments $< x^2 > = 2.0$ mm$^2$, $< x'^2 > = 2.0$ mrad$^2$, $< xx' > = 1.9$ mm.mrad, then the beam has a emittance of $\varepsilon = 0.62$ mm.mrad. However, a 5% error in the determination of $< x^2 > (< x^2 > = 2.1$ mm$^2$) results in an emittance measurement error of 24% ($\varepsilon = 0.77$ mm.mrad). This limits the accuracy of least-squares fitting techniques for determining the beam emittance, and in the worst case, can result in physically invalid (imaginary) emittance predictions. Emittance measurement algorithms used in the context of autonomous accelerator operations need to be robust to these potential errors and produce only physically valid predictions of the beam emittance.

In this work, we introduce and demonstrate a "turn-key" technique for robust, autonomous characterization of beam emittances with calibrated uncertainty estimates that requires little to no operator oversight. Our method uses a model-based algorithm, built from scratch, to autonomously choose quadrupole focusing strengths that maximize information gain about the beam size response. We then use robust statistical regression techniques to fit experimental measurements of the beam size as a function of quadrupole strength, taking into account beam dynamics principles and measurement noise. Samples drawn from the statistical model are then used to produce a detailed probability distribution of possible emittance values. This technique is demonstrated in an experiment conducted at the Argonne Wakefield Accelerator.

## 2. Materials and Methods

Here we detail our algorithm for sampling beam sizes at different quadrupole strengths and analyzing beam size data.

### 2.1. Conducting Autonomous Beamsize Measurements

Our algorithm for selecting the quadrupole strengths at which we measure the beam size is an adaptation of the Bayesian optimization [2] algorithm. Bayesian optimization starts by creating a statistical model of an objective function, known as a Gaussian process [3] (GP), to make predictions of the mean function value and corresponding uncertainty using previously measured data and expected function smoothness. This model is then used by an acquisition function to forecast the anticipated value of making future measurements. The acquisition function is then maximized to select the next parameter setting to measure.

Instead of optimizing the objective function, our algorithm, coined *Bayesian exploration* [4], aims to characterize the objective function (in this case the beam size) as a function of quadrupole strengths by choosing measurements that have the highest predicted uncertainty. This process is shown in Figure 1. Given a set of previous measurements $\{\mathbf{x}, \mathbf{y}\}$ of the RMS beam size, a GP model produces both a prediction of the beam size ($\mu(x)$) as a function of quadrupole focusing strength and the corresponding uncertainty ($\sigma(x)$) of that prediction. The acquisition function is defined as $\alpha(x) = \sigma(x)$ and is maximized to select the next value of $x$ to be observed. This, in turn, causes the sampling algorithm to choose points that maximize model uncertainty, thus maximally increasing the information gained about the beam size dependence for each experimental measurement. In one-dimensional problems, such as quadrupole scans, this algorithm will sample points in a quasi-grid like pattern, depending on the distribution of initial sample points.

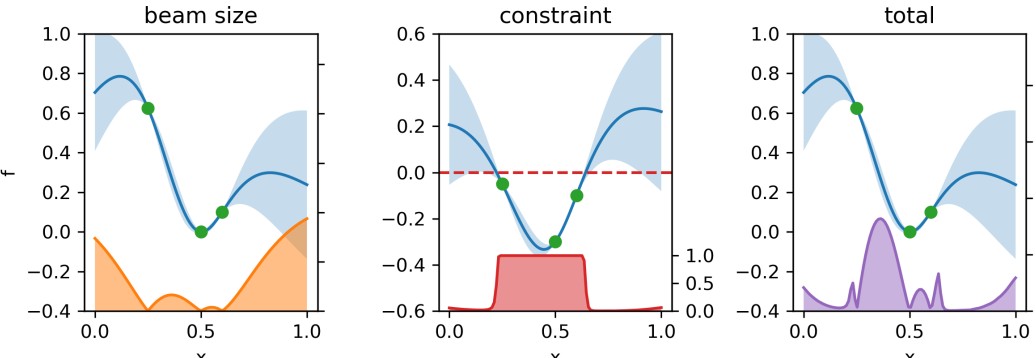

**Figure 1.** Illustration of our algorithm for characterizing unknown functions. (**Left**): Our algorithm chooses points by maximizing a acquisition function (orange) that is equal to the model uncertainty (light blue) based on previous observations (green). (**Center**): This acquisition function is then multiplied by the probability that a constraint is satisfied (red) predicted by a second, independent model using measurements of the constraining function. In this case, the constraint is satisfied when the constraining function value is less than zero. (**Right**): The total acquisition function (purple) comprised of both uncertainty and constraint terms.

In addition to this intelligent sampling strategy, our algorithm also considers observational constraints that need to be satisfied during characterization. For quadrupole scans, primary constraints involve ensuring valid beam size measurements by keeping the beam within a region of interest on the diagnostic screen and ensuring that the beam is focused enough to be discernible from background noise. As a result, the range of quadrupole strengths that result in valid beam size measurements is strongly dependent on upstream beam parameters and beamline configuration. Bayesian exploration prevents the selection of invalid quadrupole parameters by building independent GP models of each constraining function and using them to predict the likelihood that a given quadrupole strength satisfies the constraints. This process is shown in Figure 1. We determine the likelihood of an input point meeting the constraint by integrating the GP model's predicted probability distribution over constraint-compliant values. The acquisition function is scaled by this likelihood, which lessens the chance of selecting future measurements with a low probability of satisfying the constraint.

We developed a specific constraining function to effectively reduce the frequency of invalid beam size measurements in the context of imaging diagnostics. For GP models to effectively predict where input points satisfy the given constraints, the constraining functions must have a relatively smooth dependence on input parameters. To satisfy this requirement, we developed what we will refer to here as a "bounding-box" constraint, as shown in Figure 2. We specify a circular region of interest (ROI) in screen images with a center pixel coordinate $\mathbf{C}$ and a radius $r$ (also given in pixels). After processing the raw screen image of a beam (using a Gaussian smoothing filter and a fixed minimum threshold), we calculate the weighted centroid and RMS size of the beam intensity inside the ROI in both the vertical and horizontal directions. We then create a rectangular bounding box centered at the beam centroid with side lengths equal to four times the RMS beam sizes in each direction, which encapsulates most if not all of the beam intensity on the screen for observed beams. The constraint function is then defined by the maximum distance between the ROI center and the bounding box corners, $c = \mathbf{max}_i \|\mathbf{C} - \mathbf{S}_i\| - r$, where $\mathbf{S}_i$ denotes the pixel coordinates of each bounding box corner. If the beam bounding box is inside the circular ROI, then this constraining function is negative; conversely, if it extends beyond the bounding box boundary, then the constraining function value is positive. To prevent diffuse beams we use a constraint on the total intensity of all pixel values inside the ROI, requiring a minimum intensity for valid beam size measurements. If individual measurements of the beam do not satisfy all of these constraints, the measurement of the beam size is discarded while constraining function values are retained, as shown in Figure 3.

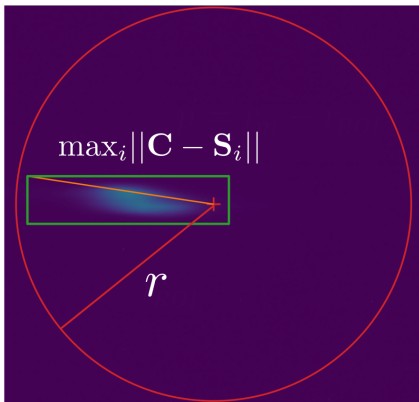

**Figure 2.** Diagram showing bounding box style image constraint.

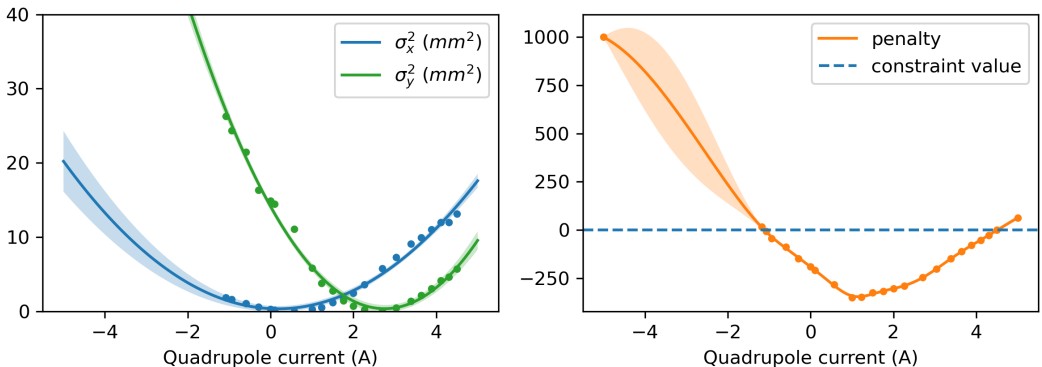

**Figure 3.** (**Left**): Beam sizes squared plotted for both the horizontal and vertical directions as a function of quadrupole current, along with Bayesian model fits using a second-order polynomial kernel. (**Right**): Constraining function measurements and model predictions. Dashed line denotes the maximum allowed value of the penalty function that satisfies the constraint. A majority of measurements chosen by our algorithm satisfy the constraint, which would not be the case for random or grid-like sampling of quadrupole strengths.

### 2.2. Calculating Emittances

Once beam size data has been collected, we determine the distribution of possible emittances from the data by drawing samples from a GP trained on the data set combined with a physics-informed kernel function. It is known from first-order beam dynamics that the beam size squared should have a quadratic dependence on the focusing strength of the quadrupole, so in turn, we use a second-order polynomial kernel for the GP model. As a result, samples drawn from the GP model will also have quadratic dependence on the quadrupole strength (see Figure 4). A corresponding emittance value for each sample is calculated by fitting each functional sample independently to the analytical model of beam transport through a quadrupole and drift, resulting in a distribution of emittance values from the GP model of the beam sizes. GP samples that predict negative beam sizes or imaginary emittances are dropped from the distribution in a process known as rejection sampling.

### 2.3. Experimental Demonstration

We conducted an experimental demonstration of automatic emittance measurements at the Argonne Wakefield Accelerator (AWA) [5]. Our study attempted to characterize the beam emittance of beams exiting the accelerating section of the AWA beamline using a single quadrupole magnet (effective length 0.12 m) and a YAG diagnostic screen located 1.065 m downstream. First, the beam was centered on the screen and manipulated by upstream quadrupoles to fit within the ROI. Then, we used the python library Xopt [6] to sample four chosen points to create an initial data set. Xopt was then used to perform constrained Bayesian exploration as described in the previous sections with a Gaussian

process. After a fixed number of iterations, the algorithm was terminated and the data was used to calculate a distribution of possible emittances.

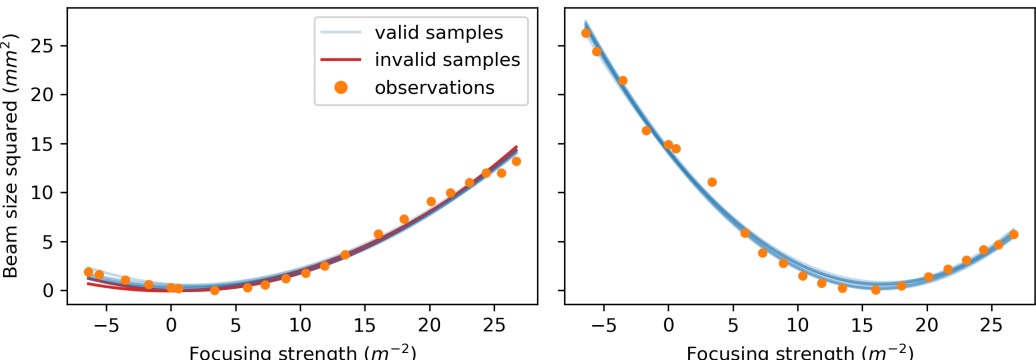

**Figure 4.** Plots showing samples drawn from the Gaussian process model used to determine the beam emittance for the horizontal (**left**) and vertical (**right**) axes. Samples that correspond to imaginary emittances are denoted as "invalid" and are rejected when calculating the distribution of predicted emittances.

## 3. Results

Results from the experimental demonstration are shown in Figures 3–5. In Figure 3 (left), we observe that Bayesian exploration distributed beam size measurements evenly throughout the valid input space of quadrupole strengths. Figure 3 (right) shows that the constraining function was learned during the exploration process, resulting in only three measurements that violated the constraint.

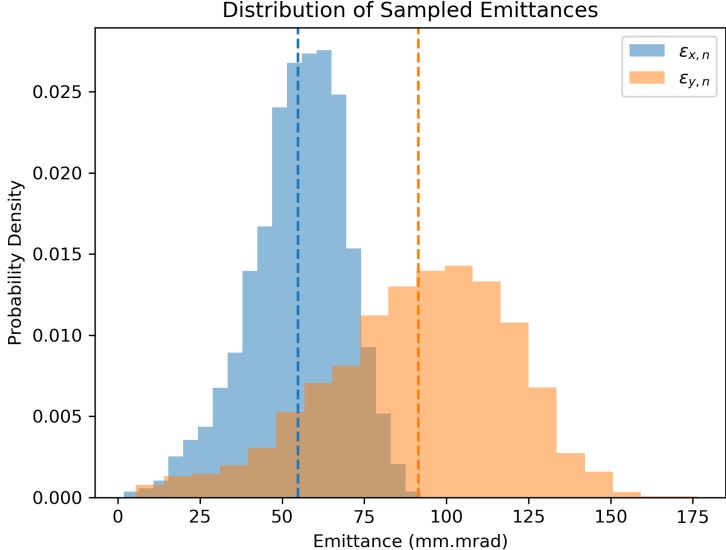

**Figure 5.** Distribution of horizontal and vertical emittances calculated using samples drawn from the predictive model. Dashed vertical lines denote mean values.

Figure 4 shows samples drawn from the GP model. An emittance value is calculated for each sample using a second-order polynomial fit to calculate elements of the beam matrix. Samples that predict an imaginary beam emittance (approximately 10% for the data sets shown here) are considered "invalid" and are rejected. Figure 5 shows predictions of the beam emittance from valid samples drawn from the predictive beamsize model. Our algorithm predicted a horizontal emittance of $\varepsilon_{x,n} = 54 \pm 15$ mm.mrad and a vertical emittance of $\varepsilon_{y,n} = 91 \pm 28$ mm.mrad. Furthermore, our algorithm identified the asymmetry in the probability distribution, with longer tails below the median value. Finally, our algo-

rithm predictions are consistent with conventional least-squares fitting of the experimental data, which predicts transverse emittances of $\varepsilon_{x,n} = 40$ mm.mrad and $\varepsilon_{y,n} = 65$ mm.mrad.

Despite relatively small uncertainty in the predictive beamsize model, there is still significant uncertainty in the beam emittance. It is likely that these large uncertainties are a result of a combination of factors, including the noisy measurements and catastrophic cancellation effects described earlier. The well-calibrated uncertainty metrics produced by our algorithm can be used to inform optimization algorithms for tuning upstream beamline parameters.

## 4. Discussion

Our results show that this algorithm is successful in automating the quadrupole scan process given arbitrary upstream beamline parameters, thus reducing the burden on accelerator operators when emittance measurements are needed. The algorithm can select quadrupole strengths to rotate the beam in phase space while adhering to practical constraints that provide valid beam size measurements. This enables future attempts to automate the optimization of beam emittances at AWA and other accelerator facilities.

This method can be further improved through several means. First, beam size measurements at every shot can be used in creating the predictive model, as opposed to using averaged measurements, which would improve the accuracy of uncertainty estimates of the emittance due to jitter. Second, the speed of decision making in the algorithm could be increased by using a mesh numerical optimizer of the acquisition function, since the decision space is only one-dimensional. Third, to promote efficient sampling of quadrupole strengths on each side of the beam size minimum, the upper confidence bound acquisition function [7] can be used with a large $\beta$ parameter to bias exploration towards quadrupole strengths that are closer to the observed beam size minimum. Finally, instead of using the beam images to calculate RMS beam sizes for fitting a polynomial model, the entire image can be used to accurately reconstruct the transverse phase space distribution, as is done in [8].

**Author Contributions:** Conceptualization, R.R. and A.E.; Data curation, R.R.; Formal analysis, R.R.; Funding acquisition, A.E.; Investigation, R.R., S.K. and E.W.; Methodology, R.R., D.K. and A.E.; Software, R.R. and D.K.; Supervision, A.E. and J.P.; Validation, R.R.; Visualization, R.R.; Writing—original draft, R.R., D.K. and A.E.; Writing—review and editing, R.R., D.K. and A.E. All authors have read and agreed to the published version of the manuscript.

**Funding:** This work was funded by the U.S. Department of Energy, Office of Science, Office of Basic Energy Sciences under Contract No. DE-AC02-76SF00515.

**Data Availability Statement:** Experimental data can be made available upon reasonable request. The algorithms used in this study are freely available as part of the Xopt package https://github.com/ChristopherMayes/Xopt (v1.4.1).

**Conflicts of Interest:** The authors declare no conflict of interest.

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
