# Peer review of "Demonstration of Autonomous Emittance Characterization at the Argonne Wakefield Accelerator"

_instruments, doi:10.3390/instruments7030029_

Round 1

Reviewer 1 Report

The paper discusses a new, automated approach to measuring emittance with a quad scan, which is a well-known and common technique in the accelerator field (though Wiedeman's book is not an appropriate reference since he really doesn't discuss it).

I think the proposed and demonstrated scheme might be quite valuable, but, to me, the material is not presented in a clear and understandable manner and I would like the paper to be rewritten with more details and clarity. (The paper may have enough details for a specialist in the AI/ML field, but the paper has too much jargon and makes too many assumptions for the majority of readers who would want to know about this, accelerator physicists like me - and if I'm having trouble understanding the paper, they will to). I have collected the majority of my comments in the following three areas.

1. Second introductory paragraph - I'm not sure I agree with the comments that implementing a quad scan is hard from a practical sense. Typically, a waist is formed at the screen and then the quad is (or quads are, if two are used) varied in strength to see the transverse RMS beam sizes change on the screen.  That's pretty much what is done here too, right? Except that the acquisition function helps reduce the number of quad strength steps used for the reconstruction?

Even though its individual phrases may be correct, I'm pretty sure I don't like the statement: "fitting noisy experimental data to simplified, analytical  models of beam transport can often result in non-physical results (imaginary emittances) and be subject to numerical errors due to catastrophic cancellation . . . " because I don't think it accurately captures what most accelerator physicists see as limitations of quad scans and this statement can be greatly improved. To be clear, "noisy experimental data" arises because it's hard to determine clean measurements of RMS beam sizes due to baselines, linearities, etc, right? I'm assuming this proposed technique also relies on "simplified, analytic models of beam transport", right? I think we exquisitely understand analytic models of beam transport (to multiple orders), though space-charge effects are hard to include since they vary for each different settings of the focusing quad. So is that the issue being referred to? And "catastrophic cancellation" would seem to refer to the case where one is taking the inverse of a matrix with two rows (or columns) that are linear combinations of others, which isn't obvious to me how it occurs in a traditional quad scan.

2. Materials and Methods - What does "target function" refer to? The beam distribution at the location of the screen? And is the function more than the six transverse beam values <xx>, <xx'>, <x'x'>, <yy>, <yy'>, and <y'y'>? I think that is what is meant in section 2.1. Also "posterior" means at the location of the screen? And the "acquisition function" is used to determine the quad strengths for the measurements?

I read up a bit on Gaussian Processes (the authors should know the link on reference 3 is wrong; also GP is used by never defined), but I am unable to understand exactly what is being done. Section 2.1 needs to be expanded to include more details. From the text, it is not clear to me if/how an analytic model for the beam transport is used, although "sampled parabolas" would indicate they are.

3. How does the emittance calculated this way agree with the standard approach? It would be nice to see a comparison with the results from a quad scan done the usual way - I'd imagine the emittance values would be the same, but maybe the RMS uncertainty would be refined by this technique?

Author Response

We appreciate the reviewers comments which have improved the quality of our paper. Our point by point responses are written below in green.

The paper discusses a new, automated approach to measuring emittance with a quad scan, which is a well-known and common technique in the accelerator field (though Wiedeman's book is not an appropriate reference since he really doesn't discuss it). 

I think the proposed and demonstrated scheme might be quite valuable, but, to me, the material is not presented in a clear and understandable manner and I would like the paper to be rewritten with more details and clarity. (The paper may have enough details for a specialist in the AI/ML field, but the paper has too much jargon and makes too many assumptions for the majority of readers who would want to know about this, accelerator physicists like me - and if I'm having trouble understanding the paper, they will to). I have collected the majority of my comments in the following three areas. 

Thank you for the insightful comments and outside perspective. We have rewritten most of the paper to reduce jargon and make the ideas more understandable to a general audience. 

  1. Second introductory paragraph - I'm not sure I agree with the comments that implementing a quad scan is hard from a practical sense. Typically, a waist is formed at the screen and then the quad is (or quads are, if two are used) varied in strength to see the transverse RMS beam sizes change on the screen.  That's pretty much what is done here too, right? Except that the acquisition function helps reduce the number of quad strength steps used for the reconstruction?

The focus of the algorithm proposed in this work is to automate the process of conducting the quadrupole scan in a way that reduces the oversight needed for valid emittance measurements. While the process of determining the fixed parameter scan is straightforward, it is time consuming and requires operator oversight. This is one of the main practical barriers faced by accelerator operators when trying to optimize beam emittance with respect to upstream parameters.

The paper text has been altered substantially to reduce jargon and clarify the advantages of our algorithm over conventional methods. 

Even though its individual phrases may be correct, I'm pretty sure I don't like the statement: "fitting noisy experimental data to simplified, analytical  models of beam transport can often result in non-physical results (imaginary emittances) and be subject to numerical errors due to catastrophic cancellation . . . " because I don't think it accurately captures what most accelerator physicists see as limitations of quad scans and this statement can be greatly improved. To be clear, "noisy experimental data" arises because it's hard to determine clean measurements of RMS beam sizes due to baselines, linearities, etc, right? I'm assuming this proposed technique also relies on "simplified, analytic models of beam transport", right? I think we exquisitely understand analytic models of beam transport (to multiple orders), though space-charge effects are hard to include since they vary for each different settings of the focusing quad. So is that the issue being referred to? And "catastrophic cancellation" would seem to refer to the case where one is taking the inverse of a matrix with two rows (or columns) that are linear combinations of others, which isn't obvious to me how it occurs in a traditional quad scan. 

It is indeed true that both previous algorithms and this work both use simplified models of beam dynamics to determine beam emittances. The verbiage that you reference has been removed from the text. On the other hand, catastrophic cancellation is a significant problem when computing beam emittances. Text has been added to the manuscript to clarify this fact, along with a numerical example that shows how small errors in reconstructing individual elements of the beam matrix can balloon into large errors when calculating emittances. 

  1. Materials and Methods - What does "target function" refer to? The beam distribution at the location of the screen? And is the function more than the six transverse beam values <xx>, <xx'>, <x'x'>, <yy>, <yy'>, and <y'y'>? I think that is what is meant in section 2.1. Also "posterior" means at the location of the screen? And the "acquisition function" is used to determine the quad strengths for the measurements?

The target function nomenclature in the paper refers to a measurement of the beam size as a function of quadrupole focusing strength. As you highlight this is vague and needlessly abstract. We have replaced this nomenclature with explicit reference to characterizing beam size dependence on quadrupole strength to make it clearer to the reader. 

I read up a bit on Gaussian Processes (the authors should know the link on reference 3 is wrong; also GP is used by never defined), but I am unable to understand exactly what is being done. Section 2.1 needs to be expanded to include more details. From the text, it is not clear to me if/how an analytic model for the beam transport is used, although "sampled parabolas" would indicate they are. 

Text has been added to Section 2.2 to clarify how analytical beam transport models are used to determine beam emittances using sampled parabolas from the GP model of the beam size. The link in reference 3 has been fixed. 

  1. How does the emittance calculated this way agree with the standard approach? It would be nice to see a comparison with the results from a quad scan done the usual way - I'd imagine the emittance values would be the same, but maybe the RMS uncertainty would be refined by this technique?

Our calculation agrees with conventional analysis methods. Prediction of the emittance from conventional least squares fitting of the data has been included in the results section for comparison. We have also included a short discussion of how our method provides a detailed probability distribution of the beam emittance as opposed to simple scalar metrics of the emittance uncertainty. 

Reviewer 2 Report

This paper addresses a very common problem in accelerator commissioning and comes up with a novel and brilliant solution to measuring emittance using artificial intelligence to guide the process so that decent results are always obtained. Time will tell if there are some systems that can defeat this technique, but they do show results with one accelerator that provides reasonable results. Future work will hopefully reduce the uncertainty of the emittance measured, but the current results are close to those derived from the current, very time-consuming procedures. Also, many times, the current procedures get no answer at all, so the procedure described here would be a major improvement.

Author Response

We appreciate the reviewers positive report and suggestions of future avenues for development.